# Optimizing Intermediate Representations of Generative Models for Phase Retrieval

**Tobias Uelwer**\*  
*Department of Computer Science*  
*Technical University of Dortmund*
*tobias.uelwer@tu-dortmund.de*

**Sebastian Konietzny**\*  
*Department of Computer Science*  
*Technical University of Dortmund*
*sebastian.konietzny@tu-dortmund.de*

**Stefan Harmeling**  
*Department of Computer Science*  
*Technical University of Dortmund*
*stefan.harmeling@tu-dortmund.de*

**Reviewed on OpenReview:** *https://openreview.net/forum?id=YAVE6jfeJb*

## Abstract

Phase retrieval is the problem of reconstructing images from magnitude-only measurements. In many real-world applications the problem is underdetermined. When training data is available, generative models allow optimization in a lower-dimensional latent space, hereby constraining the solution set to those images that can be synthesized by the generative model. However, not all possible solutions are within the range of the generator. Instead, they are represented with some error. To reduce this representation error in the context of phase retrieval, we first leverage a novel variation of intermediate layer optimization (ILO) to extend the range of the generator while still producing images consistent with the training data. Second, we introduce new initialization schemes that further improve the quality of the reconstruction. With extensive experiments on the Fourier phase retrieval problem and thorough ablation studies, we can show the benefits of our modified ILO and the new initialization schemes. Additionally, we analyze the performance of our approach on the Gaussian phase retrieval problem.

## 1 Introduction and Related Work

Optimizing intermediate layers in generative models can lead to excellent results in various linear inverse problems like inpainting, super-resolution, denoising and compressed sensing, as shown by Daras et al. [8]. In our paper, we show that their approach can be extended to solve the more difficult non-linear inverse problem of phase retrieval. In particular, we also tackle the challenging problem of non-oversampled phase retrieval, which we explain next.

For simplicity, we define different instances of the phase retrieval problem for the one-dimensional case. However, extensions to two or more dimensions are straight-forward.

**Fourier Phase Retrieval:** In many imaging applications in physics, we are only able to measure the magnitude of the Fourier transform of an image, e.g., in astronomy [10], X-ray crystallography [25] or optics [35]. In the Fourier setting the magnitude measurements $y \in \mathbb{R}^n$ are given as

$$y = |Ax|, \tag{1}$$

---

\*equal contribution

where $A = F \in \mathbb{C}^{n \times n}$ is the discrete Fourier transform (DFT) matrix and $x \in \mathbb{R}^n$ is the original image (here represented as a vector). However, due to the symmetry of the DFT, $y$ has only $\lfloor n/2 \rfloor + 1$ many distinct values and thus recovering the $n$ entries of $x$ is underdetermined. Thus we require additional information about $x$, which, in this paper, will be learned from example images to constrain the solution space.

In contrast, the common assumption in Fourier phase retrieval is that $x$ is zero-padded and thus the problem is no longer underdetermined. The zero-padding of $x$ leads to an oversampling of the relevant signal and makes the problem much easier to solve. Again, in this work, we assume that the Fourier measurements are not oversampled, i.e., we consider the case, where the image $x$ has not been zero-padded and $n$ entries need to be recovered.

**Compressive Gaussian Phase Retrieval:** A related phase retrieval problem is the compressive Gaussian phase retrieval problem (discussed in the work of Candes et al. [4] and Shechtman et al. [30]). For this problem, the DFT matrix is replaced by a matrix $A$ with randomly sampled entries (real- or complex-valued, typically sampled from a Gaussian distribution), i.e., the measurements $y \in \mathbb{R}^m$ are given by

$$y = |Ax|. \tag{2}$$

If the number of entries of $y$ does not exceed the number of entries of $x$, i.e., $A \in \mathbb{R}^{m \times n}$ has fewer rows than columns, the problem is said to be undersampled.

Note, that both variants of phase retrieval are more difficult than linear inverse problems due to the non-linearity (the absolute value) in the forward model. Both problems have received a lot of attention in the literature. In the following, we give a short overview of the various approaches distinguishing between classical optimization-based and learning-based methods.

## 1.1 Methods without Learning for Phase Retrieval

One of the first (Fourier) phase retrieval algorithms is Fienup's error-reduction (ER) algorithm [11] that is based on alternating projections. In his work, Fienup [11] also introduced the hybrid-input-output (HIO) algorithm which improved the ER algorithm. The Gaussian phase retrieval problem was approached by Candes et al. [4] who used methods based on Wirtinger derivatives to minimize a least-squares loss. Wang et al. [36] considered a similar idea but used a different loss function and so-called truncated generalized gradient iterations. Holographic phase retrieval is a related problem, which aims to reconstruct images from magnitude measurements where a known reference is assumed to be added onto the image. This problem was approached by Lawrence et al. [19] using untrained neural network priors. Untrained neural networks have also been used by Chen et al. [5] for solving different phase retrieval problems. Phase retrieval using the RED framework [29] was done by Metzler et al. [24], Wang et al. [37] and Chen et al. [6]. However, these works focus on the reconstruction of images from coded diffraction pattern measurements or $4\times$ oversampled Fourier magnitudes.

## 1.2 Learning-based Methods for Phase Retrieval

Recently, learning-based methods for solving phase retrieval problems gained a lot of momentum. They can be classified into two groups: supervised methods and unsupervised methods.

**Supervised methods:** These methods directly learn a mapping that reconstructs images from the measurements. Learning neural networks for solving Fourier phase retrieval was done by Nishizaki et al. [27]. This approach was later extended to a neural network cascade by Uelwer et al. [32], who also used conditional generative adversarial networks to solve various phase retrieval problems [33]. While giving impressive results, all of these supervised approaches have a drawback for the Gaussian phase retrieval setup, since the measurement operator must be known at training time. Supervised learning of reference images for holographic phase retrieval was done by Hyder et al. [15] and Uelwer et al. [34] gave further insights.

**Unsupervised methods:** Unsupervised methods are agnostic with respect to the measurement operator, because they are trained on a dataset of images without the corresponding measurements. Generative

models for Gaussian phase retrieval have been analyzed by Hand et al. [13] and Liu et al. [21]. Killedar & Seelamantula [18] apply a sparse prior on the latent space of the generative model to solve Gaussian phase retrieval. Alternating updates for Gaussian phase retrieval with generative models was proposed by Hyder et al. [14]. Manekar et al. [23] used a passive loss formulation to tackle the non-oversampled Fourier phase retrieval problem.

In the context of these related works, the method that we propose in this paper can be seen as an unsupervised learning-based approach that can optionally be extended with a supervised component, as we detail in Section 2.2.

### 1.3 Optimizing Representations of Generators for Inverse Problems

Generative models have been used to solve linear inverse problems, e.g., compressed sensing [3], image inpainting [41]. To decrease the representation error and thereby boost the expressiveness of these generative models, the idea of optimizing intermediate representations was successfully applied in the context of compressed sensing [31], image inpainting and super-resolution [8]. The former paper called this approach generator surgery, the latter intermediate layer optimization (ILO).

### 1.4 Our Claims and Contributions

The claims made in this paper are the following:

**Claim 1:** *We claim that the idea of optimizing intermediate representations in generative models can be applied to solve phase retrieval problems. However, given the difficulty of these non-linear inverse problems an additional subsequent optimization step is required to obtain good results.*

**Claim 2:** *We also claim that the need for multiple runs with random restarts for generator-based approaches to solve inverse problems can be reduced by using different initialization schemes.*

Further intuition and motivation will be given in Section 2.1.

In extensive experiments, we consider the underdetermined Fourier and Gaussian phase retrieval (with real and complex measurement matrices) and show that our method provides state-of-the-art results. By analyzing various combinations of methods and generator networks, we show the influence of the choice of the generator. In further ablation studies, we show the importance of each component of our method.

## 2 Phase Retrieval with Generative Models

The basic idea of applying a trained generative model $G : \mathbb{R}^l \to \mathbb{R}^n$ to the phase retrieval problem is to plug $G(z)$ into the least-squares data fitting term and to optimize over the latent variable $z$, i.e.,

$$\min_z \left\| |AG(z)| - y \right\|_2^2. \tag{3}$$

This method has been used for Gaussian phase retrieval (with real-valued $A$) by Hand et al. [13]. While this approach yields good results, the reconstruction quality is limited by the range of the generator network $G$. This issue is, for example, also discussed in the work of Asim et al. [2]. In the following, we explain how the range of $G$ can be extended by optimizing intermediate representations of $G$ instead of only solving Equation 3 with respect to the latent variable $z$.

**Notation:** For the exposition of the method we use the following notation: The generator network $G = G_k \circ \cdots \circ G_1$ can be written as the concatenation of $k$ layers $G_1, \ldots, G_k$. The subnetwork consisting of layers $i$ through $j$ is denoted by $G_i^j = G_j \circ \cdots \circ G_i$. Note, that $G = G_1^k = G_{i+1}^k \circ G_1^i$ for $1 \le i \le k$. The output of layer $G_i$ is written as $z_i$, i.e., $z_i = G_i(z_{i-1})$. The input of $G$ is $z_0$.

Furthermore, we define the $\ell_1$-ball with radius $r > 0$ around $z$ as,

$$B_r(z) = \big\{ x \mid \|x - z\|_1 \le r \big\}. \tag{4}$$

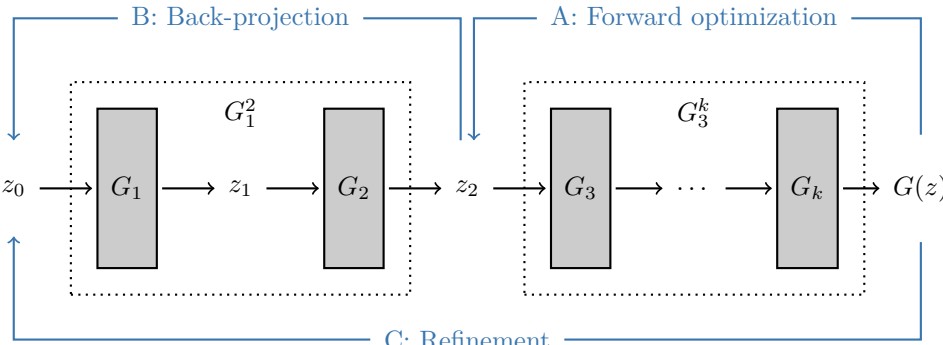

Figure 1: Structure of the generator network and, in blue, the parts relevant for steps A, B and C.

### 2.1 Phase Retrieval with Intermediate Layer Optimization (PRILO)

The key idea of ILO is to vary the intermediate representations learned by a sub-network of the generator $G$ while ensuring an overall consistency of the solution. The latter is particularly challenging for non-linear problems like phase retrieval. As we will show in the experiments, additional steps and special initialization schemes are essential to obtain useful results. All of these steps will be explained next.

The generator network captures the prior knowledge and restricts the solutions of the (underdetermined) phase retrieval problem. In this work, we use two important ideas:

**Range extension:** Hidden representations $z_i$ for $i > 0$ are allowed to vary outside the range of $G_1^i$. This will reduce the representation error of the generator.

**Image consistency:** Solutions must be realistic, non-degenerate and similar to the the training dataset. This is achieved by back-projection ensuring that $z_i$ is not too far away from the image of some $z_0$.

Note that one has to strive for a trade-off between range extension and image consistency, where range extension facilitates the optimization, while image consistency regularizes the solution.

Concretely, we repeatedly split the generator $G$ into sub-networks $G_1^i$ and $G_{i+1}^k$ and iteratively optimize the intermediate representations $z_i$ resulting from $G_1^i$ (instead of simply optimizing the initial latent variable $z_0$). In this way, we go outside the range of the sub-network $G_1^i$ to reduce the mean-squared magnitude error. To make the resulting image more consistent we back-project $z_i$ closer to the range of $G_1^i$. We will show that $z_0$ from the back-projection is not only a good candidate for the overall optimization, but also serves as a good initialization for further optimization leading to a significant refinement of the overall solution.

To get started, we initially optimize the input variable $z_0$ (minimize Equation 3). This provides starting points for all intermediate representations $z_i$ for $i > 0$. The overall procedure then iteratively applies the following three steps to different intermediate layers:

**A. Forward optimization:** vary the intermediate representation $z_i$ to minimize the magnitude error while staying in the ball with radius $r_i$ around the current value of $z_i$, i.e.,

$$z_i^* = \arg\min_{z \in B_{r_i}(z_i)} \left\| |A G_{i+1}^k(z)| - y \right\|_2^2. \tag{5}$$

Note, that this step possibly leaves the range of $G_1^i$ (range extension). By introducing ball-constraints on the optimized variable we avoid overfitting the measurements.

**B. Back-projection:** find an intermediate representation $G_1^i(z_0)$ that is close to the optimal $z_i^*$ from step A, i.e.,

$$\bar{z}_0 = \arg\min_{z \in B_{s_i}(\mathbf{0})} \left\| G_1^i(z) - z_i^* \right\|_2^2, \tag{6}$$

where $\mathbf{0} \in \mathbb{R}^l$ denotes the vector of all zeros. By doing so we ensure that there exists a latent $z_0$ that yields the reconstruction. Thereby, we regularize the solution from step A to obtain image consistency.

While these two steps have been shown to improve the overall performance of linear inverse problems, we found that for phase retrieval the following refinement step significantly improves the reconstructions.

**C. Refinement:** Our intuition here is that the back-projected latent $\bar{z}_0$ serves as a good initialization for further optimization. Starting from the $\bar{z}_0$ found in step B, we again optimize the measurement error

$$z_0^* = \underset{z \in B_{r_0}(\bar{z}_0)}{\arg\min} \left\| |AG(z)| - y \right\|_2^2. \tag{7}$$

Note, that as we only optimize the latent variable of $G$, we do not have to apply the back-projection again for this refinement step.

These steps are applied repeatedly to the different intermediate representations. Multiple strategies for selecting the layers to optimize are possible. In practice it turned out to be sufficient to treat the layers once from left to right. Thus, we recommend to start with the layers closer to the input and then progress further with the layers closer to the output. Starting with the later layers can cause problems as they offer too much flexibility and no regularizing effect. The final reconstruction is obtained as $x^* = G(z_0^*)$. Steps A, B and C are visualized in Figure 1.

The three optimization problems stated in Equations 5-7 are solved using projected gradient descent. This means that after each gradient descent step the iterate is projected back onto the appropriate $\ell_1$-ball. In our implementation this projection is implemented using the method described by Duchi et al. [9].

## 2.2 Initialization Schemes

Due to the difficulty of the phase retrieval problem, many methods are sensitive to the initialization. Candes et al. [4] proposed a spectral initialization technique that is often used for compressive Gaussian phase retrieval. Since it requires one to solve a large eigenvalue problem, this method can be quite costly. A common approach to improve reconstruction performance is to use random restarts and to select the solution with the lowest magnitude error after the optimization [13]. However, this requires running the method multiple times. Instead, we propose two fast initialization schemes that use the generative model and the available magnitude information before the optimization.

**Magnitude-Informed Initialization (MII):** Deep generative models can generate a lot of images at relatively low cost: instead of optimizing Equation 3 with random restarts, we sample a set of starting points $Z = \{z_0^{(0)}, \ldots, z_0^{(p)}\}$ and select the one with the lowest magnitude mean-squared error (MSE),

$$z_0^{\text{init}} = \underset{z \in Z}{\arg\min} \left\| |AG(z)| - y \right\|_2^2, \tag{8}$$

before the optimization. The empirical reason for this choice is that the MSE between the unknown target image and the initial reconstructions $G(z)$ for $z \in Z$ strongly correlates with the MSE of their magnitudes (correlation $\rho = 0.91$). This initialization scheme is applicable to other reconstruction algorithms that search in the latent space of a generative model as well.

**Learned Initialization (LI):** The generator is trained on a dataset of images that are characteristic for the problem. We use the generator to train an encoder network $E_\theta$ that maps magnitude measurements $y$ to latent representations $z_0$. Once the encoder is trained, we can use it to predict an initialization for the optimization discussed in Section 2.1.

A naive way to train the encoder network is to create input/output pairs by starting with random latent vectors $z_0$ and combining them with the magnitudes $|AG(z_0)|$ of the corresponding image. The disadvantage

is that the original training images are only implicitly used in this approach (because those were used to train the generator). To leverage the generator and also the training images, we estimate the weights $\theta$ of the encoder $E_\theta$, such that encoded magnitudes $E_\theta(y)$ generate an image $G(E_\theta(y))$ that is close to the original image $x$. This idea originates from GAN inversion [43].

More precisely, we minimize a combination of three loss functions to train the encoder

$$\mathcal{L}_{\mathrm{MSE}}(G(E_\theta(y)), x) + \lambda_{\mathrm{perc}}\mathcal{L}_{\mathrm{perc}}(G(E_\theta(y), x)) + \lambda_{\mathrm{adv}}\mathcal{L}_{\mathrm{adv}}(D_\phi(G(E_\theta(y)), D_\phi(x))), \tag{9}$$

where $\mathcal{L}_{\mathrm{MSE}}$ is the image MSE, $\mathcal{L}_{\mathrm{perc}}$ the LPIPS loss [42], and $\mathcal{L}_{\mathrm{adv}}$ is a Wasserstein adversarial loss [1] with gradient penalty [12] (using the discriminator network $D_\phi$). In our experiments, we set $\lambda_{\mathrm{perc}} = 5 \cdot 10^{-5}$ and $\lambda_{\mathrm{adv}} = 0.1$. Note, that the generator network is fixed and we only optimize the encoder weights $\theta$ and the discriminator weights $\phi$ to solve a learning objective.

Additionally, we also found it helpful to apply a small normally-distributed perturbation with mean 0 and standard deviation $\sigma = 0.05$ to the predicted latent representation. We do so because using a fixed initialization leads to a deterministic optimization trajectory when running the optimization multiple times.

For better exploration of the optimization landscape we also apply gradient noise during optimization. In combination with the projection onto the feasible set the update reads as

$$z^{(k+1)} = P\left(z^{(k)} - \alpha\left(\nabla_z f\left(z^{(k)}\right) + u^{(k)}\right)\right), \tag{10}$$

where $f$ is the objective function corresponding to the current step, $u^{(k)} \sim \mathcal{N}(0, \sigma_k^2 I)$ with $\sigma_k^2 = \frac{\eta}{(1+k)^\gamma}$ and $P$ is the projection onto $B_r(x^{(0)})$. In our experiments we set $\eta = 0.02$ and $\gamma = 0.55$. This noise decay schedule was proposed by Welling & Teh [39] and is also discussed by Neelakantan et al. [26] in the context of neural network training.

One drawback of the learned initialization is that it requires one to retrain the encoder network when the measurement matrix changes. In the following, we only evaluate this approach for the Fourier phase retrieval problem.

## 3 Experimental Evaluation

We evaluate our method on the following datasets: MNIST [20], EMNIST [7], FMNIST [40], and CelebA [22]. The first three datasets consist of $28 \times 28$ grayscale images, whereas the latter dataset is a collection of $200,000$ color images that we cropped and rescaled to a resolution of $64 \times 64$.

Similar to Hand et al. [13], we use a fully-connected variational autoencoder (VAE) for the MNIST-like datasets and a DCGAN [28] for the CelebA dataset. Both architectures were also used by Bora et al. [3] for compressed sensing. Details about the VAE training are described in Appendix A. Going beyond existing works, we are interested in improving the performance on the CelebA dataset even further, thus we considered deeper, more expressive generators, like the Progressive GAN [16] and the StyleGAN [17]. Since the optimization of the initial latent space for deeper generators is more difficult, we expect significantly better results by adaptively optimizing the successive layers of these models. Our implementation is based on open-source projects[1] [2] [3]. In total our computations took two weeks on two NVIDIA A100 GPU. Detailed hyperparameter settings can be found in Appendix G.

### 3.1 Phase Retrieval with Fourier Measurements

For the problem of Fourier phase retrieval, we compare our method first with the ER algorithm [11] and the HIO algorithm [11] (both having no learning component) and then with the following supervised learning methods: an end-to-end (E2E) learned multi-layer-perceptron [33], a residual network [27], a cascaded

---

[1] https://github.com/rosinality/progressive-gan-pytorch
[2] https://github.com/rosinality/style-based-gan-pytorch
[3] https://github.com/giannisdaras/ilo

Table 1: Fourier phase retrieval: Our proposed method (being unsupervised) performs best among classical, unsupervised and supervised approaches for MNIST and EMNIST, and second best for FMNIST (in terms of PSNR). Best values are printed **bold** and second-best are underlined.

| Method | Learning | MNIST | | FMNIST | | EMNIST | |
|--------|----------|-------|---|--------|---|--------|---|
| | | PSNR ($\uparrow$) | SSIM ($\uparrow$) | PSNR ($\uparrow$) | SSIM ($\uparrow$) | PSNR ($\uparrow$) | SSIM ($\uparrow$) |
| ER [11] | – | 15.1826 | 0.5504 | 12.7002 | 0.3971 | 13.1157 | 0.5056 |
| HIO [11] | – | 23.4627 | 0.5274 | 12.9530 | 0.4019 | 14.2230 | 0.4942 |
| DPR [13] | unsup. | 23.9871 | 0.9005 | 18.8657 | 0.6846 | 20.9448 | 0.8568 |
| PRILO (ours) | unsup. | 42.7584 | 0.9843 | 20.2087 | 0.7133 | 33.8263 | 0.9367 |
| PRILO-MII (ours) | unsup. | **43.9541** | **0.9949** | 21.4836 | 0.7560 | **37.1807** | **0.9719** |
| ResNet [27] | sup. | 17.0886 | 0.7292 | 17.4787 | 0.6070 | 14.4627 | 0.5616 |
| E2E [33] | sup. | 18.5041 | 0.8191 | 20.2827 | 0.7400 | 17.4049 | 0.7598 |
| CPR [32] | sup. | 19.6216 | 0.8529 | 20.9064 | 0.7768 | 19.3163 | 0.8537 |
| PRCGAN* [33] | sup. | 41.3767 | 0.9890 | **25.5513** | **0.8376** | 27.1948 | 0.9416 |

multilayer-perceptron [32] and a conditional GAN approach [33]. Additionally, we apply DPR which was originally only tested on Gaussian phase retrieval [13].

Table 1 and Table 2 show the mean peak-signal-to-noise-ratio (PSNR) and the mean structural similarity index measure (SSIM) [38]. Results with 95%-confidence intervals can be found in Appendix E. Each reported number was calculated on the reconstructions of 1024 test samples. We allow four random restarts and select the generated sample resulting in the lowest measurement error. As one can see, more expressive models result in better reconstructions. Furthermore, we note that our proposed initialization schemes, i.e., the LI and MII, lead to improved performance. Figure 2 shows typical reconstructions from the CelebA dataset. Reconstructions using the DCGAN and the Progressive GAN can be found in Appendix B and Appendix C..

Surprisingly, our unsupervised PRILO-MII approach often quantitatively outperforms the supervised competitors (MNIST and EMNIST). Only for FMNIST we get slightly worse results which we attribute to the performance of the underlying generator of the VAE. Notably, we outperform DPR that uses the same generator network, which shows that the modifications explained in this paper are beneficial. Summarizing, among classical and unsupervised methods our new approach performs best, often even better than the supervised methods.

The CelebA dataset is more challenging and the influence of the choice of the generator networks is substantial. We consider the DCGAN [28], the Progressive GAN [16] and the StyleGAN [17] as base models for our approach (second column in Table 2). Furthermore, the new initialization schemes have a strong impact. For fairness, we also combined the existing DPR approach [13] with the various generator models and the initialization schemes. Thus we are able to study the influence of the different components. Among the generators, StyleGAN performs best. MII improves the results, while LI achieves even better reconstructions. For all previously mentioned combinations, our new method PRILO is better than DPR.

To further support our claims, we performed additional experiments on the high-resolution FFHQ dataset [17] using the StyleGAN architecture. The results can be found in Appendix F. Although, the reconstruction capabilities of all considered methods are still limited, our proposed changes improve the reconstructions.

## 3.2 Phase Retrieval with Gaussian Measurements

Beyond Fourier phase retrieval, compressed Gaussian phase retrieval is a similarly challenging problem, where the measurement matrix $A$ has real- or complex-valued normally-distributed entries. For the real-valued case, we sample from a zero-mean Gaussian with variance $1/m$,

$$A = (a_{kl})_{\substack{k=1,\ldots,m \\ l=1,\ldots,n}} \sim \mathcal{N}\left(0, \frac{1}{m}\right), \tag{11}$$

Table 2: Fourier phase retrieval on CelebA: Our proposed PRILO combined with LI and StyleGAN achieves the best results in terms of PSNR and SSIM, also against enhanced version of DPR, which uses LI and StyleGAN. Best values are printed **bold** and second-best are underlined.

| | | | CelebA | |
|---|---|---|---|---|
| Method | Base model | Learning | PSNR (↑) | SSIM (↑) |
| ER [11] | – | – | 10.4036 | 0.0637 |
| HIO [11] | – | – | 10.4443 | 0.0510 |
| DPR [13] | DCGAN | unsup. | 16.9384 | 0.4457 |
| DPR-MII | DCGAN | unsup. | 17.8651 | 0.4898 |
| PRILO | DCGAN | unsup. | 18.3597 | 0.5159 |
| PRILO-MII | DCGAN | unsup. | 18.5656 | 0.5264 |
| DPR | Progressive GAN | unsup. | 18.4384 | 0.5276 |
| DPR-MII | Progressive GAN | unsup. | 19.5213 | 0.5738 |
| PRILO | Progressive GAN | unsup. | 19.2779 | 0.5665 |
| PRILO-MII | Progressive GAN | unsup. | 19.9057 | 0.5910 |
| DPR | StyleGAN | unsup. | 18.2606 | 0.4859 |
| DPR-MII | StyleGAN | unsup. | 20.3889 | 0.5972 |
| PRILO | StyleGAN | unsup. | 18.5542 | 0.5143 |
| PRILO-MII | StyleGAN | unsup. | 21.4223 | 0.6358 |
| E2E [33] | – | sup. | 20.1266 | 0.6367 |
| PRCGAN* [33] | – | sup. | 22.1951 | 0.6846 |
| DPR-LI | StyleGAN | sup. | 22.6223 | 0.7021 |
| PRILO-LI | StyleGAN | sup. | **23.3378** | **0.7247** |

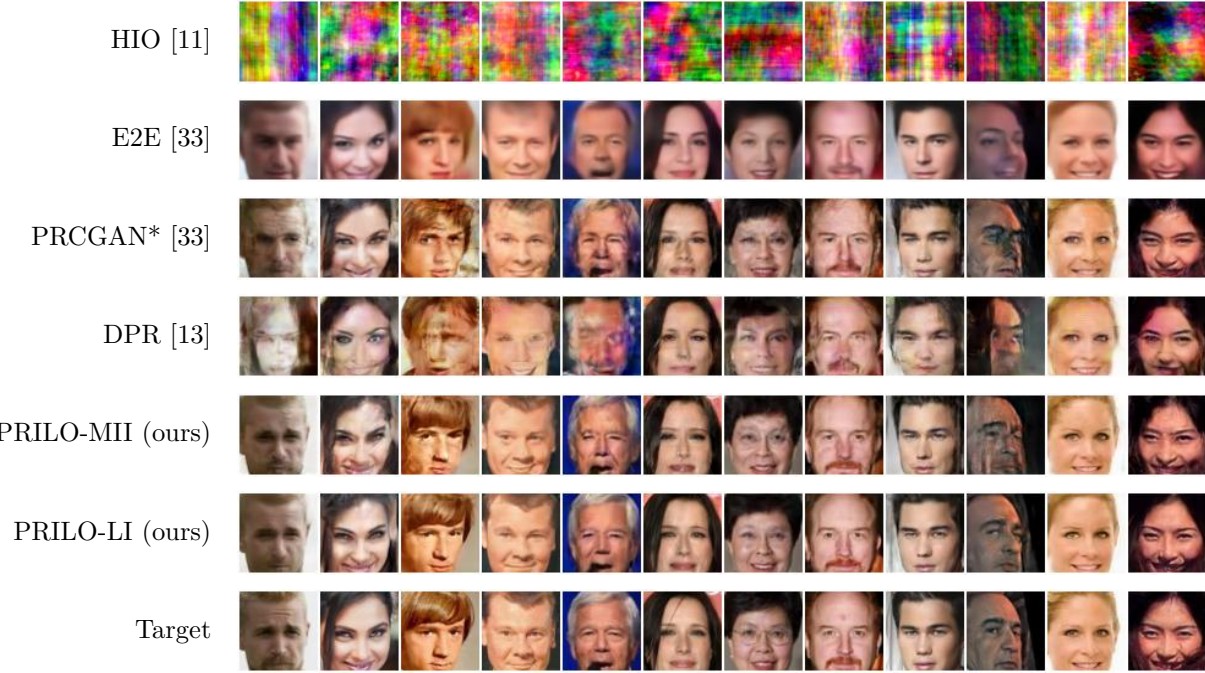

Figure 2: Fourier measurements on CelebA: only PRILO-LI (based on StyleGAN) is able to reconstruct finer details of the faces and produces realistic images with few artifacts.

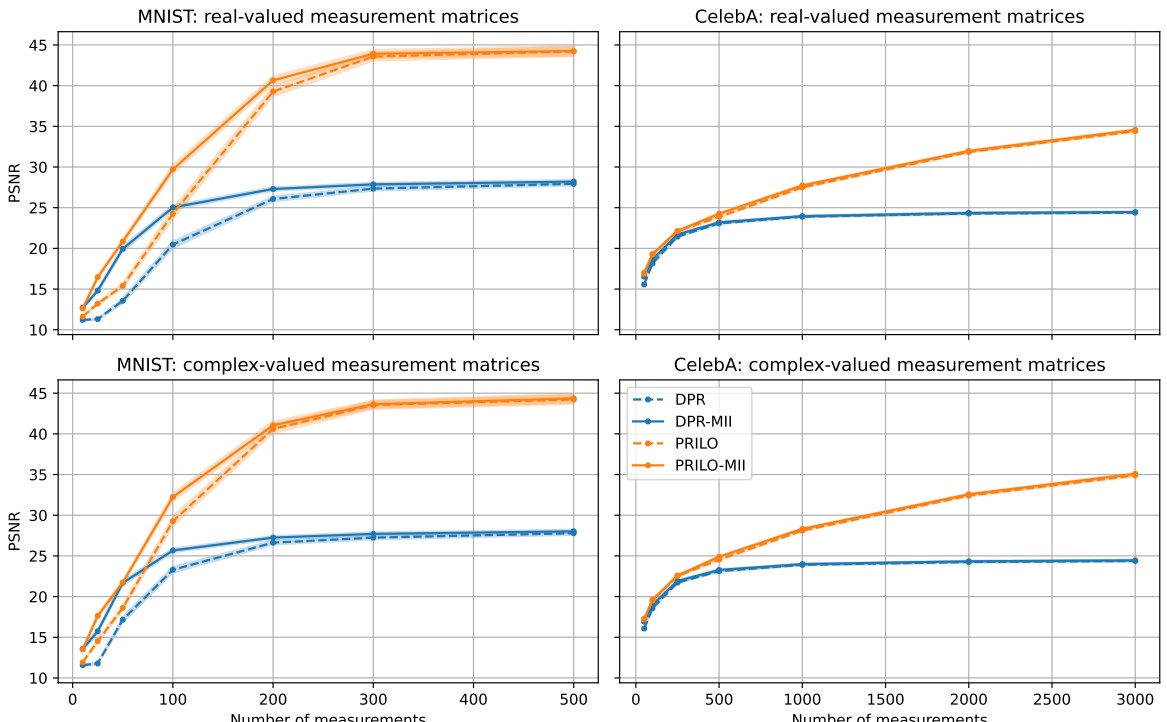

Figure 3: Real and complex Gaussian phase retrieval on MNIST and CelebA: PRILO-MII greatly improves DPR's reconstruction results. Note, that we also combine DPR with our MII. For MNIST we used the VAE and for CelebA the StyleGAN as base model.

where $m$ is the number of measurements. The entries of the complex-valued measurement matrix are obtained by sampling real and imaginary parts from a zero-mean Gaussian with variance $1/(2m)$,

$$A = (a_{kl} + ib_{kl})_{\substack{k=1,\ldots,m \\ l=1,\ldots,n}} \text{ with } a_{kl}, b_{kl} \sim \mathcal{N}\left(0, \frac{1}{2m}\right). \tag{12}$$

We directly compare our results with the results of the DPR method for real- and complex-valued measurement matrices (the real-valued case was already covered in [13]). Figure 3 shows the PSNR and the SSIM values for different numbers of measurements $m$. Our proposed method PRILO-MII (orange) improves DPR's results (blue) by a margin. Note that for compressive Gaussian phase retrieval only unsupervised methods are tested, since each measurement matrix requires retraining the model.

## 3.3 Ablation Studies

In our experiments, we observed that the performance can be improved by re-running the optimization procedure with a different initialization and selecting the result with the lowest magnitude error which is a common practice in image reconstruction. However, this approach is quite costly. Our initialization schemes described in Section 2.2 can be used to achieve a similar effect without the need of re-running the whole optimization procedure multiple times. In Figure 4, we compare the single randomly initialized latent variable, 5000 initialization using the MII approach and slightly perturbed learned initialization (LI). Again, we consider a test set of 1024 samples. Although, we are still using restarts in combination with our initialization, we observe that already for a single run our initialization outperforms the results of 5 runs.

In order to assess the impact of each step on the reconstruction performance, we perform an ablation study. We compare our complete PRILO-MII and PRILO-LI models with different modified versions of the model: for one, we consider omitting every optimization, i.e., we only compare with the initialization. Next, we omit the back-projection step (B) and the refinement step (C). We also compare with the variant of our model

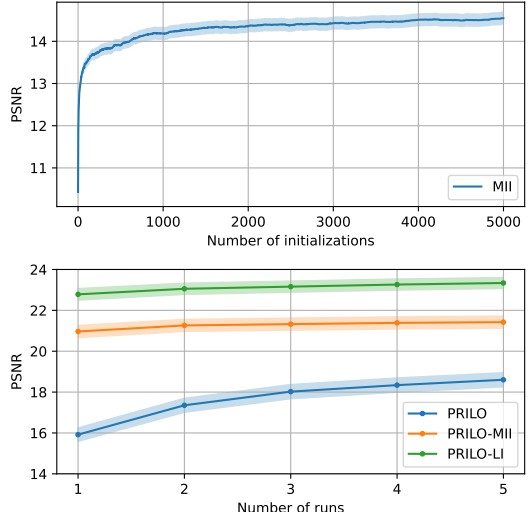

Figure 4: Effect of the number of initializations and number of restarts onto the reconstruction performance. Shaded areas indicate the 95%-confidence interval.

Table 3: Ablation experiments using StyleGAN on CelebA for Fourier measurements.

| Method | PSNR (↑) | SSIM (↑) |
| --- | --- | --- |
| PRILO-LI | 23.3378 | 0.7247 |
|    only initialization | 16.9070 | 0.4777 |
|    only step A | 23.0097 | 0.7108 |
|    only step A and B | 22.9109 | 0.7063 |
|    no ball constraint | 22.8192 | 0.6974 |
| PRILO-MII | 21.4223 | 0.6358 |
|    only initialization | 15.0566 | 0.3565 |
|    only step A | 20.1736 | 0.5861 |
|    only step A and B | 19.9409 | 0.5719 |
|    no ball constraint | 19.8461 | 0.5540 |

which we only omit step C. Finally, we analyze the impact of the ball-constraints. Table 3 shows that each of the components is important to reach the results. Notably, only using step A and B (which corresponds to naïvely applying the approach by Daras et al. [8] to phase retrieval) gives worse results than only step A. However, in combination with our proposed step C the approach gives better results. Figure 7 in Appendix D shows how omitting different steps affects the optimization.

## 4 Conclusion

Generative models play an important role in solving inverse problem. In the context of phase retrieval, we observe that optimizing intermediate representations is essential to obtain excellent reconstructions. Our method PRILO, used in combination with our new initialization schemes, produces better reconstruction results for Gaussian and Fourier phase retrieval than existing (supervised and unsupervised) methods. Notably, in some cases our unsupervised variant PRILO-MII even outperforms existing supervised methods. We also show that the initialization schemes we introduced can easily be adapted to different methods for inverse problems that are based on generative models, e.g., DPR. Our ablation study justifies that each of our used components is essential to achieve the reported results.

**Limitations**

During our experiments, we observed that some hyperparameter tuning is necessary to achieve good results: the selected intermediate layer highly influences the results and also the radii for the constraints play an important role. Furthermore, our approach is in some cases not able to reconstruct the finer details and is still (to some extend) limited by the generative model (FMNIST results reported in Table 1). Furthermore, generator-based methods for non-oversampled Fourier phase retrieval still struggle to reconstruct high-resolution images as demonstrated in Appendix F.

**Broader Impact Statement**

This work discusses a method for reconstructing images from their magnitude measurements. Whether this method can have negative societal impacts is difficult to predict and depends on the specific application, e.g.,

in X-ray crystallography or microscopy. For example, in some applications the reconstructed images could contain sensitive information, which could lead to data security issues if the method is not applied properly.

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

## A   VAE Architecture

For the MNIST, FMNIST and EMNIST dataset we use a variational autoencoder (VAE) as a generator. The encoder and the decoder each consist of three layers with ReLU activation function, where each layer had 500 hidden units. The latent space was chosen 100-dimensional. It was trained using a Bernoulli-likelihood and Adam with learning rate $10^{-3}$ for 100 epochs.

## B   CelebA Reconstructions using the DCGAN

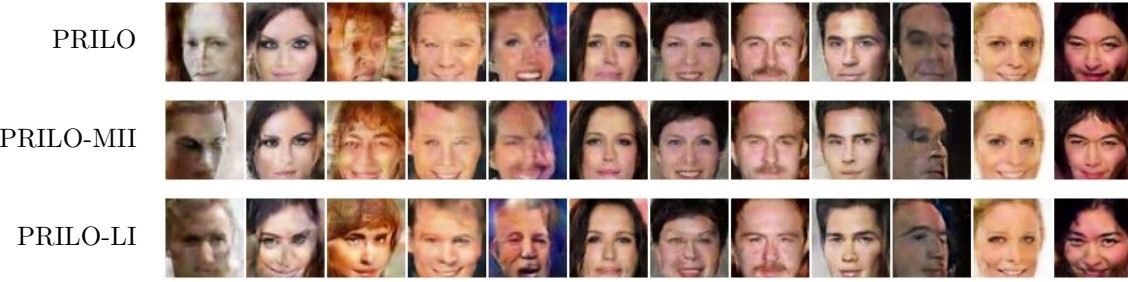

Figure 5: Reconstructions from Fourier measurements on CelebA by PRILO and PRILO-MII based on the DCGAN [28].

## C   CelebA Reconstructions using the Progressive GAN

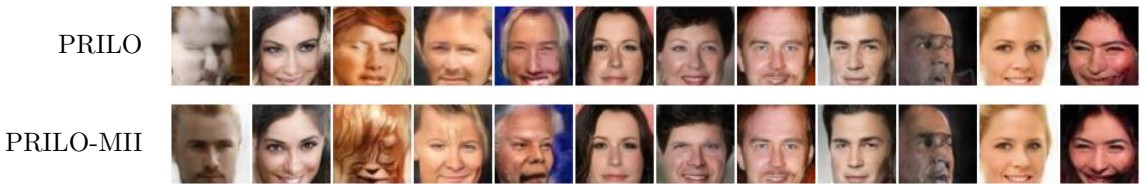

Figure 6: Reconstructions from Fourier measurements on CelebA by PRILO and PRILO-MII based on the Progressive GAN [16].

## D   Ablation Experiments: PSNR Throughout the Optimization

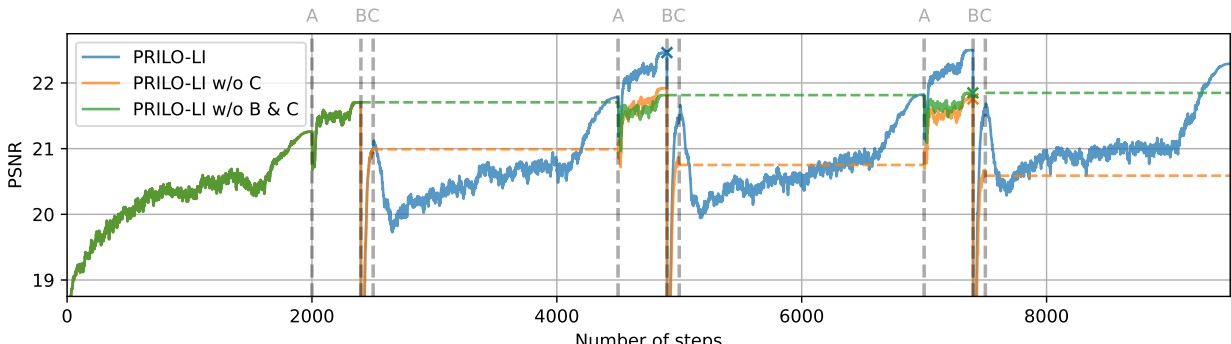

Figure 7: Ablation experiments for PRILO-LI based on StyleGAN on CelebA data: We omit different steps of our method and report the PSNR on a validation set consisting of 64 images. PSNR corresponding to the lowest magnitude error are highlighted.

# E Quantitative Results with Confidence Intervals

## E.1 MNIST, FMNIST and EMNIST

Table 4: Fourier phase retrieval: Our proposed method (being unsupervised) performs best among classical, unsupervised and supervised approaches for MNIST and EMNIST, and second best for FMNIST (in terms of PSNR). Best values are printed **bold** and second-best are underlined. Numbers printed after ± are 95%-confidence intervals calculated on a batch of 1024 images.

| Method | Learning | MNIST | | FMNIST | | EMNIST | |
| --- | --- | --- | --- | --- | --- | --- | --- |
| | | PSNR (↑) | SSIM (↑) | PSNR (↑) | SSIM (↑) | PSNR (↑) | SSIM (↑) |
| ER | – | 15.1826 ± 0.3782 | 0.5504 ± 0.0122 | 12.7002 ± 0.1960 | 0.3971 ± 0.0122 | 13.1157 ± 0.2767 | 0.5056 ± 0.0113 |
| HIO | – | 20.5696 ± 1.9740 | 0.5274 ± 0.0130 | 12.9530 ± 0.2002 | 0.4019 ± 0.0117 | 14.1934 ± 0.4872 | 0.4942 ± 0.0113 |
| DPR | unsup. | 24.2118 ± 0.3407 | 0.9070 ± 0.0071 | 18.8657 ± 0.2434 | 0.6846 ± 0.0113 | 20.9448 ± 0.3219 | 0.8568 ± 0.0097 |
| PRILO (ours) | unsup. | 42.7584 ± 0.5617 | 0.9843 ± 0.0042 | 20.2087 ± 0.3300 | 0.7133 ± 0.0120 | 33.8263 ± 0.6440 | 0.9367 ± 0.0091 |
| PRILO-MII (ours) | unsup. | **43.9541 ± 0.4574** | **0.9949 ± 0.0019** | 21.4836 ± 0.3364 | 0.7560 ± 0.0111 | **37.1807 ± 0.5257** | **0.9719 ± 0.0060** |
| ResNet | sup. | 17.0886 ± 0.1838 | 0.7292 ± 0.0082 | 17.4787 ± 0.1730 | 0.6070 ± 0.0109 | 14.4627 ± 0.1614 | 0.5616 ± 0.0100 |
| E2E | sup. | 18.5041 ± 0.1509 | 0.8191 ± 0.0059 | 20.2827 ± 0.2165 | 0.7400 ± 0.0095 | 17.4049 ± 0.1756 | 0.7598 ± 0.0084 |
| CPR | sup. | 19.6216 ± 0.1528 | 0.8529 ± 0.0047 | 20.9064 ± 0.2236 | 0.7768 ± 0.0090 | 19.3163 ± 0.1772 | 0.8537 ± 0.0057 |
| PRCGAN-L | sup. | 41.3767 ± 0.5584 | 0.9890 ± 0.0022 | **25.5513 ± 0.4457** | **0.8376 ± 0.0098** | 27.1948 ± 0.3873 | 0.9416 ± 0.0067 |

## E.2 CelebA

Table 5: Fourier phase retrieval on CelebA: Our proposed PRILO combined with LI and StyleGAN achieves the best results in terms of PSNR and SSIM, also against enhanced version of DPR, which uses LI and StyleGAN. Best values are printed **bold** and second-best are underlined. Numbers printed after ± are 95%-confidence intervals calculated on a batch of 1024 images.

| | | | CelebA | |
|---|---|---|---|---|
| Method | Base model | Learning | PSNR (↑) | SSIM (↑) |
| ER | – | – | $10.4036 \pm 0.1282$ | $0.0637 \pm 0.0028$ |
| HIO | – | – | $10.4443 \pm 0.1271$ | $0.0510 \pm 0.0023$ |
| DPR | DCGAN | unsup. | $16.9384 \pm 0.2534$ | $0.4457 \pm 0.0124$ |
| DPR-MII | DCGAN | unsup. | $17.8651 \pm 0.2485$ | $0.4898 \pm 0.0123$ |
| PRILO | DCGAN | unsup. | $18.3597 \pm 0.2781$ | $0.5159 \pm 0.0136$ |
| PRILO-MII | DCGAN | unsup. | $18.5656 \pm 0.2431$ | $0.5264 \pm 0.0119$ |
| DPR | Progressive GAN | unsup. | $18.4384 \pm 0.3091$ | $0.5276 \pm 0.0147$ |
| DPR-MII | Progressive GAN | unsup. | $19.5213 \pm 0.2942$ | $0.5738 \pm 0.0141$ |
| PRILO | Progressive GAN | unsup. | $19.2779 \pm 0.3040$ | $0.5665 \pm 0.0141$ |
| PRILO-MII | Progressive GAN | unsup. | $19.9057 \pm 0.2894$ | $0.5910 \pm 0.0137$ |
| DPR | StyleGAN | unsup. | $18.2606 \pm 0.3745$ | $0.4859 \pm 0.0181$ |
| DPR-MII | StyleGAN | unsup. | $20.3889 \pm 0.3279$ | $0.5972 \pm 0.0154$ |
| PRILO | StyleGAN | unsup. | $18.5542 \pm 0.3435$ | $0.5143 \pm 0.0165$ |
| PRILO-MII | StyleGAN | unsup. | $21.4223 \pm 0.2973$ | $0.6358 \pm 0.0135$ |
| E2E | – | sup. | $20.1266 \pm 0.1731$ | $0.6367 \pm 0.0082$ |
| PRCGAN-L | – | sup. | $22.1951 \pm 0.2248$ | $0.6846 \pm 0.0097$ |
| DPR-LI | StyleGAN | sup. | $22.6223$ $\pm 0.2455$ | $0.7021$ $\pm 0.0112$ |
| PRILO-LI | StyleGAN | sup. | **$23.3378$** $\pm 0.2691$ | **$0.7247$** $\pm 0.0120$ |

# F High-resolution Images

Table 6: Fourier phase retrieval on high-resolution images from FFHQ using a StyleGAN. Best values are printed **bold** and second-best are underlined. Numbers printed after ± are 95%-confidence intervals calculated on a batch of 128 images.

| | | | CelebA | |
|---|---|---|---|---|
| Resolution | Method | Learning | PSNR (↑) | SSIM (↑) |
| $256 \times 256$ | DPR | unsup. | $12.8724 \pm 0.7167$ | $0.2942 \pm 0.0257$ |
| | DPR-MII | unsup. | **$14.4224$** $\pm 0.6878$ | $0.3468$ $\pm0.0270$ |
| | PRILO | unsup. | $12.9225 \pm 0.6873$ | $0.3066 \pm 0.0252$ |
| | PRILO-MII | unsup. | $14.3080$ $\pm 0.7276$ | **$0.3722$** $\pm0.0282$ |
| $512 \times 512$ | DPR | unsup. | $12.9071 \pm 0.6832$ | $0.3673 \pm 0.0232$ |
| | DPR-MII | unsup. | $14.4423$ $\pm 0.6957$ | $0.4129$ $\pm 0.0223$ |
| | PRILO | unsup. | $12.9824 \pm 0.6939$ | $0.3812 \pm 0.0237$ |
| | PRILO-MII | unsup. | **$14.5264$** $\pm 0.6907$ | **$0.4562$** $\pm 0.0236$ |
| $1024 \times 1024$ | DPR | unsup. | $12.3979 \pm 0.6383$ | $0.3891 \pm 0.0193$ |
| | DPR-MII | unsup. | $13.7563$ $\pm 0.6514$ | $0.4888$ $\pm 0.0201$ |
| | PRILO | unsup. | $12.3938 \pm 0.6429$ | $0.3953 \pm 0.0188$ |
| | PRILO-MII | unsup. | **$13.8730$** $\pm 0.6577$ | **$0.4993$** $\pm 0.0197$ |

DPR

DPR-MII

PRILO

PRILO-MII

Target

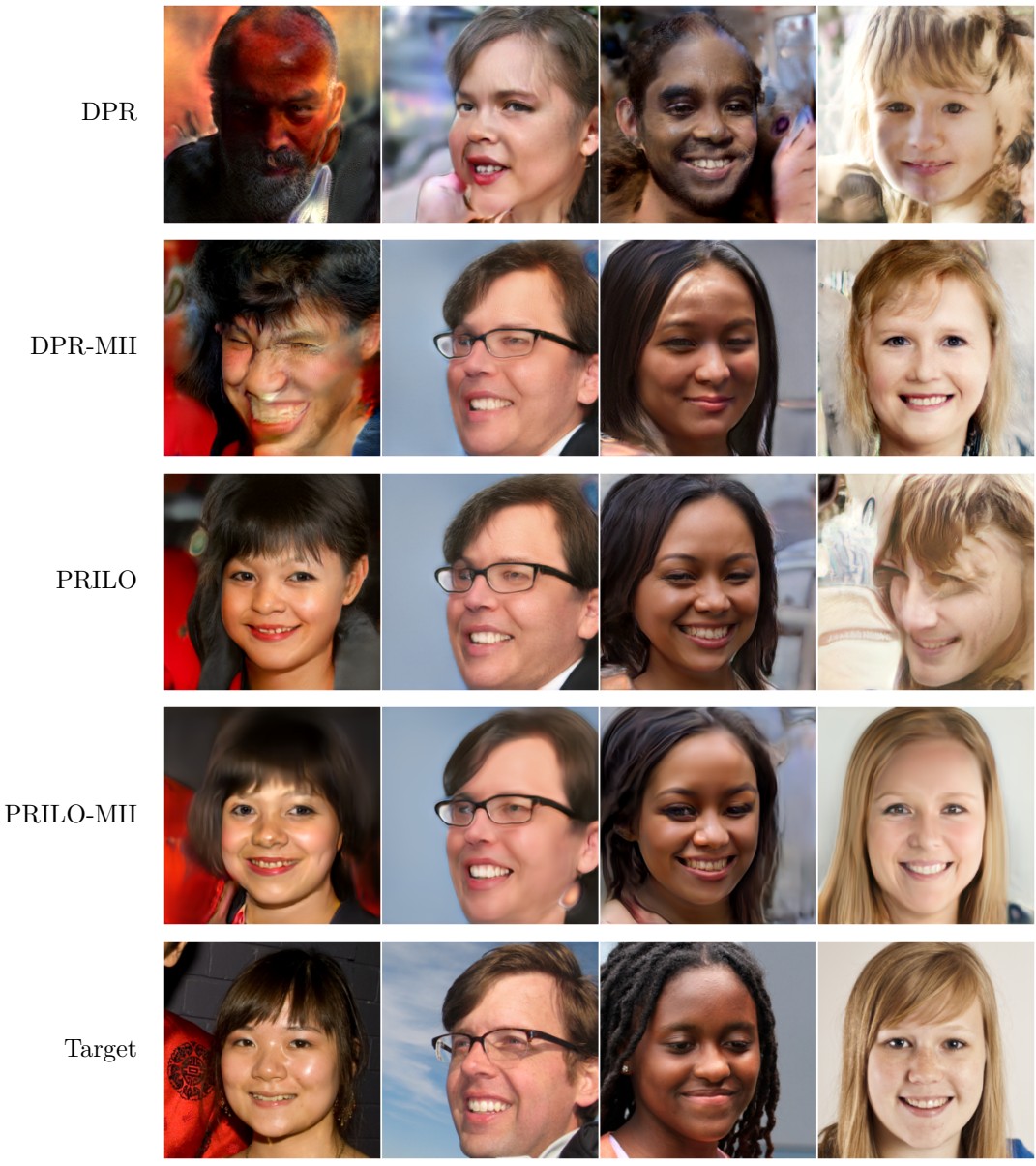

Figure 8: $512 \times 512$ reconstructions from Fourier measurements on FFHQ by DPR, DPR-MII, PRILO and PRILO-MII based on the StyleGAN.

## G  Hyperparameter Selection

Table 7: Hyperparameters for PRILO and PRILO-MII on the MNIST and EMNIST datasets.

| Repetitions | | Variable | Steps | $\ell_1$-radius |
|---|---|---|---|---|
| 1 | Initial optimization | $z_0$ | 150 | 100 |
| 1 | Forward optimization | $z_1$ | 150 | 50 |
| | Back-projection | $z_0$ | - | - |
| | Refinement | $z_0$ | - | - |
| 9 | Forward optimization | $z_2$ | 300 | 100 |
| | Back-projection | $z_0$ | - | - |
| | Refinement | $z_0$ | - | - |

Table 8: Hyperparameters for PRILO and PRILO-MII on the Fashion-MNIST dataset.

| Repetitions | | Variable | Steps | $\ell_1$-radius |
|---|---|---|---|---|
| 1 | Initial optimization | $z_0$ | 100 | 100 |
| 1 | Forward optimization | $z_1$ | 100 | 10 |
| | Back-projection | $z_0$ | - | - |
| | Refinement | $z_0$ | - | - |
| 4 | Forward optimization | $z_2$ | 200 | 40 |
| | Back-projection | $z_0$ | - | - |
| | Refinement | $z_0$ | - | - |

Table 9: Hyperparameters for PRILO (based on StyleGAN) on the CelebA dataset. Note that $z_0$ is passed into each layer. That is why we have to apply $\ell_1$-regularization also for $z_0$ when optimizing the latter layers.

| Repetitions | | Variable | Steps | $\ell_1$-radius ($z_i$) | $\ell_1$-radius ($z_0$) |
|---|---|---|---|---|---|
| 1 | Initial optimization | $z_0$ | 100 | 1000 | 1000 |
| 1 | Forward optimization | $z_1$ | 100 | 200 | 100 |
| | Back-projection | $z_0$ | 100 | 1000 | 1000 |
| | Refinement | $z_0$ | 100 | 1000 | 1000 |
| 1 | Forward optimization | $z_2$ | 100 | 200 | 100 |
| | Back-projection | $z_0$ | 100 | 1000 | 1000 |
| | Refinement | $z_0$ | 100 | 1000 | 1000 |

Table 10: Hyperparameters for PRILO-MII and PRILO-LI (based on StyleGAN) on the CelebA dataset. Note that $z_0$ is passed into each layer. That is why we have to apply $\ell_1$-regularization also for $z_0$ when optimizing the latter layers.

| Repetitions | | Variable | Steps | $\ell_1$-radius $(z_i)$ | $\ell_1$-radius $(z_0)$ |
|:---:|---|:---:|:---:|:---:|:---:|
| 1 | Initial optimization | $z_0$ | 2000 | 1000 | 1000 |
| 3 | Forward optimization | $z_1$ | 400 | 100 | 100 |
| | Back-projection | $z_0$ | 100 | 1000 | 1000 |
| | Refinement | $z_0$ | 2000 | 1000 | 1000 |

