# OpenReview forum: "Optimizing Intermediate Representations of Generative Models for Phase Retrieval"
_TMLR — Accepted by TMLR_

### Review · Reviewer_8SrU · 2022-09-11

**Summary Of Contributions:**

# Summary
## Background
The paper focuses on the phase retrieval problem, a specific nonlinear inverse problem with various applications from crystallography to coherent diffraction imaging (CDI), and leverages deep generative models with intermediate layer optimization (ILO) to solve this inverse problem. Briefly, given only the Fourier magnitudes, the goal is to reconstruct an image (i.e., retrieve the phase); however, this problem is ill-posed, as the Fourier magnitudes do not uniquely identify the phase. A possible solution is to learn image priors and constrain the solution space to the learned priors, which one can do with generative models. Therefore, to solve the inverse problem, one can optimize over the domain (i.e., the latent space) of a generative model to generate an image in the range of the model with Fourier magnitudes that match the observation.
## Contributions
In this work, the authors show that by leveraging intermediate layer optimization (ILO), one can significantly improve this process and the quality of reconstructed images. In particular, the authors propose an iterative process involving three steps. In the first step, the authors perform a constrained optimization on the intermediate layers (constrained to a $\ell_1$-ball), which they call forward optimization. In the second step, they back-project the optimized intermediate representation to update the latent code in the input domain of the generator. These two steps have also been used in prior work. However, the authors propose a third step, in which they refine the latent code through another constrained optimization. Importantly, inverse problem optimization is highly non-convex, and a good initialization plays a vital role in obtaining a satisfactory performance. To that end, the authors propose two alternative initialization schemes, namely Magnitude Informed Initialization (MII) and Learned Initialization (LI), through a learned encoder (similar to GAN inversion models). Lastly, the proposed approach is tested on Fourier and Gaussian phase retrieval problems on MNIST, FashionMNIST, EMNIST, and CelebA datasets showing improved performance over existing methods.


**Broader Impact Concerns:**

The broader impact is sufficiently discussed by the authors.

**Requested Changes:**

* There are several implementation details that I missed in the paper.
  * How is the intermediate layer, i.e., layer $i$, selected to perform the optimization? Is this done sequentially?
  * The three steps proposed in the paper could be applied iteratively and on different layers. What is the effect of the choice of the sequence of layers to optimize?

Including these details in the paper could further increase the paper's impact.

**Strengths And Weaknesses:**


## Strengths

* The paper is very well written, and the ideas are presented in a clear manner
* The paper addresses a classic problem that is of interest to various imaging applications with modern generative models
* The experiments clearly delineate the contributions of the proposed steps.  Also, the effect of the proposed initializations is tested not only on the proposed method, PRILO, but also on the DPR method [Hands et al. 2018]  showing a consistent improvement in performance.
* The experiments were conducted on various neural architectures showing consistent gains independent of the architecture choice
* The experiments were conducted with both supervised and unsupervised learned priors

## Weaknesses

While I do not see major weaknesses in this paper, below are a few points that stood out to me.

* While the experiments in the paper are great, they are on datasets that do not reflect the actual applications of phase retrieval. Importantly, the proposed approach relies on the assumption of having a good generative model of the images of interest. However, constructing such generative models in applications relevant to phase retrieval (e.g., crystallography or CDI) is not trivial.

* Minor Typo(s):
  * Page 3 - "Equation equation 3"

## Overall Assessment

To summarize my review, I think the paper is well written, and it discusses an important problem that is of interest to many imaging applications. I did not find any technical flaws in the paper, and the paper's experimental section supports the authors' claims. Overall, this is a good paper with sufficient technical contributions, nice experimental section, and a simple and direct narrative.

---

> ### Author Response · Authors · 2022-09-30
> **Response to reviewer 8SrU**
>
> We thank Reviewer 8SrU for their valuable time and their feedback! In the following we address the concerns pointed out in the review:
>
> > While the experiments in the paper are great, they are on datasets that do not reflect the actual applications of phase retrieval. Importantly, the proposed approach relies on the assumption of having a good generative model of the images of interest. However, constructing such generative models in applications relevant to phase retrieval (e.g., crystallography or CDI) is not trivial.
>
> We agree that a practical application of this method is not trivial and requires more work as the generative model needs to be tailored to the specific domain of interest. However, focusing on natural images allows easier reproducibility of the experiments, as is common in the phase retrieval and compressed sensing literature [3,12,13,14,17,19].
>
> > How is the intermediate layer, i.e., layer i, selected to perform the optimization? Is this done sequentially?
>
> Applying PRILO to the first intermediate representations behaves similar to the vanilla latent optimization (i.e., similar to DPR). That is why we start with the first layers and progress further to the deeper layers. The other way around could causes problems: starting by optimizing the deeper layers first offers too much flexibility and does not yield meaningful reconstructions (although the magnitude error might be low). We added this description to the paper (see page 4).
>
> > The three steps proposed in the paper could be applied iteratively and on different layers. What is the effect of the choice of the sequence of layers to optimize?
>
> PRILO worked best when being applied first to the layers closer to the latent variable and only later to the layers closer to the output of the generator. We repeated the steps multiple times for the same layer. We added the sequence in which we optimized the layer for each dataset in the appendix.
>
> > Minor Typo(s)
>
> We fixed the typos in the current revision of the paper. Thank you for pointing them out!

---

### Review · Reviewer_iiKt · 2022-09-17

**Summary Of Contributions:**

This work proposes a generative model based approach for phase retrieval. Specifically, the authors design an intermediate layer optimization approach and introduce an initialization scheme that further improves the data reconstruction quality. Extensive empirical results show the effectiveness of the approach on certain datasets.

**Broader Impact Concerns:**

This work proposes an approach to reconstructing data based on their magnitudes. In many applications, the reconstructed data could contain sensitive information. There could be a negative societal impact (e.g., data security issue) if the approach is not used properly. It would be good to expand the discussion in the "Broader Impact Statement" section more.

**Requested Changes:**

1. One major concern is the technical novelty of the work. The proposed approach relies heavily on feature engineering and parameter tuning. There is very limited novelty on the technical side and it also remains unclear where the performance boost comes from. It would be good to provide some extra discussion and theoretical insights. In terms of the experiment, it would be good to report averaged results using different random seeds. Since the contribution is mainly on the empirical side, it would be good to also consider higher-resolution image datasets.

2. The proposed model framework is similar to a hierarchical generative model. It would be good to add some discussion with hierarchical generative models (e.g., hierarchical VAE).

3. The parameters should be formally defined. For instance, in eq (4), it would be good to define $r$ (e.g., a positive scalar) first. In section 2, should it be $G_1^{k}=G_{i+1}^{k}\circ G_{1}^{i}$?

4. At the same time, section 1.4 seems to be tenuous. Instead of having two claims, it would be better to provide more intuition or analysis to convince the readers why that is the case.

5. It would be important to provide more analysis and elaborate on why eq (9) is a good objective function. It also seems that the performance will depend heavily on the selection of the weighting constant $\lambda$. It would be good to elaborate on how you select the weights and provide some theoretical analysis (if possible). How robust the weights are for different datasets?

6. It would be better to also include stronger and more recent unsupervised baselines in Table 1.



**Strengths And Weaknesses:**

**Strength:**
1. The paper is relatively well-written.
2. This paper considers an important and interesting application of generative models.
3. The ablation studies are detailed.

**Weakness:**
1. Limited theoretical insights. The proposed approach focuses more on the engineering side (e.g., feature engineering, combining different training losses). The approach is not theoretically justified and appears to rely heavily on parameter tuning.
2. In section 1.4, there are only claims but without analysis, insights, or evidence. The arguments can be tenuous without proper justification.
3. The training objective in equation (9) is not properly justified in the phrase retrieval context. It also appears similar to the GAN inversion objective. It would be good to explicitly explain why each term is necessary, why it is a valid objective function and how you select the weight $\lambda$. At the same time, the learning objective seems to be hand-crafted without too many theoretical insights.
4. Need to compare with stronger baselines. It would be good to compare with more recent (learning-based) unsupervised baselines in Table 1. As pointed out in 3.1, baseline [9] and [10] do not have learning components. So it is not surprising that the proposed learning-based method can easily outperform the two baselines.
5. For the empirical results, it would be good to report the error rates (e.g., averaged performance over multiple runs/random seeds).
6. The image datasets considered here are relatively low-resolution. It would be good to consider higher-resolution datasets.

---

> ### Author Response · Authors · 2022-09-30
> **Response to reviewer iiKt**
>
> We would like to thank reviewer iiKt for their detailed feedback and their valuable time! Please, consider our responses to the raised points:
>
> > 1. One major concern is the technical novelty of the work. The proposed approach relies heavily on feature engineering and parameter tuning. There is very limited novelty on the technical side and it also remains unclear where the performance boost comes from. It would be good to provide some extra discussion and theoretical insights. In terms of the experiment, it would be good to report averaged results using different random seeds. Since the contribution is mainly on the empirical side, it would be good to also consider higher-resolution image datasets.
>
> - Technical novelty: Our paper describes an application (and modification, see claims 1 and 2) of an existing method to a different problem (as invited in the TMLR submission guidelines). Our extensive ablation studies give insights about the ILO approach in the setting of phase retrieval, while it was originally only applied to linear inverse problems.
> - Relies heavily on feature engineering and parameter tuning: Our approach does **not** rely on feature engineering and the amount of hyperparameter tuning was reasonable.
> - Unclear where the performance boost comes from: We perform ablation experiments that answer exactly this question: step B and C are necessary to achieve the better results.
> - Averaged results: Note, that the results reported are calculated on a batch of 1024 samples (not only on the images shown in the work). We included a Table 1 with standard errors (calculated on these 1024 images) in the appendix. We argue that the large sample size yields statistical meaningful results and we would did not experience any changes in the results when running the experiments with a different seed.
> - Higher resolution: We also applied DPR and PRILO using a StyleGAN trained on 256x256, 512x512 and 1024x1024 FFHQ images. Results are added to Appendix F. We were able to show that our proposed modifications improve the reconstructions, however the overall performance of generator-based methods for non-oversampled Fourier phase retrieval is still limited. We are not aware of any works that considers higher-resolution images for non-oversampled Fourier phase retrieval.
>
> > 2. The proposed model framework is similar to a hierarchical generative model. It would be good to add some discussion with hierarchical generative models (e.g., hierarchical VAE).
>
> Please note, that our work can be applied to any generative model, including hierarchical ones (like the progressive GAN or the StyleGAN considered in our work) and non-hierarchical ones (like the DCGAN or the VAE used for the MNIST-like datasets). We added a remark in the second paragraph of Section 3 to avoid confusion.
>
> > 3. The parameters should be formally defined. For instance, in eq (4), it would be good to define $r$ (e.g., a positive scalar) first. In section 2, should it be $G = G_1^k=  G_{i+1}^k \circ G_1^i$.
>
> We fixed the typo and defined the radius $r>0$. Thanks for pointing these misstakes out! We also fixed Equation (10) and the Equation references.
>
> > 4. At the same time, section 1.4 seems to be tenuous. Instead of having two claims, it would be better to provide more intuition or analysis to convince the readers why that is the case.
>
> We added a sentence in Section 1.4 to direct the reader to Section 2.1, where we give intuition about why intermediate layer optimization works. Forward optimization (step A) promotes range extension, whereas back-projection (step B) promotes image consistency. The resulting iterate can be further improved by the additional refinement step as we show in Table 3.

---

> ### Author Response · Authors · 2022-09-30
> **Response to reviewer iiKt (continued)**
>
> > 5. It would be important to provide more analysis and elaborate on why eq (9) is a good objective function. It also seems that the performance will depend heavily on the selection of the weighting constant $\lambda$. It would be good to elaborate on how you select the weights and provide some theoretical analysis (if possible). How robust the weights are for different datasets?
>
> No, the choice of the different $\lambda$ only has a minor influence on the reconstruction. In fact, we used the same values as the original work [38]. No hyperparameter tuning was necessary at this point! The combination of pixel-wise, adversarial and perceptual loss functions is a very common choice in image reconstruction, see also [40,41,42,43,44] just to mention a few.
>
> > 6. It would be better to also include stronger and more recent unsupervised baselines in Table 1.
>
> Note, that DPR was originally only applied to Gaussian phase retrieval in combination with a DCGAN prior, i.e., we applied DPR+DCGAN, DPR+Progessive GAN and DPR+StyleGAN to the Fourier phase retrieval problem to obtain meaningful baselines. Also DPR in combination with MII and LI are included for fair comparison. We argue that these baselines are the **most appropriate baselines to verify our claims.** We contacted Killedar et al. [45] and asked for their code, however they did not respond yet. Please, let us know whether you insist on additional baselines.
>
> > Broader Impact Concerns
>
> We added your point to our broader impact statement. Thanks for pointing it out!
>
>
> ### REFERENCES:
>
> [40] Sajjadi, Mehdi SM, Bernhard Schölkopf, and Michael Hirsch. "Enhancenet: Single image super-resolution through automated texture synthesis." Proceedings of the IEEE international conference on computer vision. 2017.
>
> [41] Zhang, Xuaner, Ren Ng, and Qifeng Chen. "Single image reflection separation with perceptual losses." Proceedings of the IEEE conference on computer vision and pattern recognition. 2018.
>
> [42] Shen, Ziyi, et al. "Deep semantic face deblurring." Proceedings of the IEEE conference on computer vision and pattern recognition. 2018.
>
> [43] Liu, Ziyang, et al. "DSRGAN: Detail Prior-Assisted Perceptual Single Image Super-Resolution via Generative Adversarial Networks." IEEE Transactions on Circuits and Systems for Video Technology (2022).
>
> [44] Liang, Jie, Hui Zeng, and Lei Zhang. "Details or Artifacts: A Locally Discriminative Learning Approach to Realistic Image Super-Resolution." Proceedings of the IEEE/CVF Conference on Computer Vision and Pattern Recognition. 2022.
>
> [45] Killedar, Vinayak, and Chandra Sekhar Seelamantula. "Compressive Phase Retrieval Based On Sparse Latent Generative Priors." ICASSP 2022-2022 IEEE International Conference on Acoustics, Speech and Signal Processing (ICASSP). IEEE, 2022.

---

### Review · Reviewer_J5zR · 2022-09-24

**Summary Of Contributions:**

The paper proposes a method for phase retrieval by (1) optimizing intermediate representations (modifying intermediate layer optimization, ILO, method), named PRILO, and (2) two initialization schemes. To validate the method, the paper uses two examples of Fourier phase retrieval and compressive Gaussian phase retrieval as an implicit application, and shows improves generated image quality as the performance measure. The proposed method does not require additional human supervision, thus unsupervised method, and improves the quality of generated images in noticeable gains (large gain in MNIST variant datasets, meaningful gain in CelebA dataset).

**Broader Impact Concerns:**

No much concerns here.

**Requested Changes:**

- In abstract, "generative models are a new idea to constrain the solution set" is not clear. Specifically, how does the generative model constrain the solution set?
- In 2.1. B. Back-projection section, why this serves as a regularizer for the solution by step A?
	- Isn't z_0 in this section \bar{z_0}?
- The reviewer encourages to empirically compare the "random restarts and to select the solution with the lowest magnitude error after the optimization" (in Sec. 2.2) to the proposed method.
	- Likely in Figure 4
	- Also citation is needed for this sentence
- Please explain why you only evaluate LI initialization on Fourier phase retrieval.
- Please explain why why step C (the proposed method) is not very effective in StyleGAN on CelebA for Fourier measurements (Table 3 - PRILO-LI section)
- In Table 1, the latest method compared is DPR [12], which is published in 2018. We have list of relevant work as follows to name a few:
	- Kiledar et al., Compressive Phase Retrieval Based On Sparse Latent Generative Priors, ICASSP 2022
	- Chen et al., Unsupervised Phase Retrieval Using Deep Approximate MMSE Estimation, IEEE Transactions on Signal Processing, April 2022
	- Chen et al., Phase Recovery With Deep Complex-Domain Priors, IEEE Signal Processing Letters, March 2022
- In all equation reference, "Equation equation ??" should be "Equation ??"
- Figure 1 in text (C. Refinement in Sec. 2.1) has no link.

**Strengths And Weaknesses:**

Strengths
- S1: Meaningful gain in quality of generated images without additional supervision
- S2: Proposed method is simple yet quite effective

Weaknesses
- W1: The proposed method is for phase retrieval but there is no analysis of the retrieved phases, instead the paper provides indirect analysis of the results -- evaluating the quality of generated images (probably via retrieved phases)
- W2: While the method is proposed for both Fourier and Gaussian phase retrieval, most of the analyses are done in Fourier phase retrieval (except the Fig 3). Is there anything special in this method working better in Fourier phase retrieval?
- W3: Some presentations are not clear (see requested changes)

---

> ### Author Response · Authors · 2022-09-30
> **Response to reviewer J5zR**
>
> We thank reviewer J5zR for their insightful comments and their valuable time! In the following we would like to address these points:
>
> > W1: The proposed method is for phase retrieval but there is no analysis of the retrieved phases, instead the paper provides indirect analysis of the results -- evaluating the quality of generated images (probably via retrieved phases)
>
> While analysis of the retrieved phases could be interesting, in practical applications one is usually only interested in reconstructing the images, e.g. electron densities.
>
> > W2: While the method is proposed for both Fourier and Gaussian phase retrieval, most of the analyses are done in Fourier phase retrieval (except the Fig 3). Is there anything special in this method working better in Fourier phase retrieval?
>
> The Fourier phase retrieval problem is the problem that is more relevant in practice and more difficult to solve (in the non-oversampled case). Compressive Gaussian phase retrieval is a theoretical toy problem that we included for completeness and to compare against DPR.
>
> > In abstract, "generative models are a new idea to constrain the solution set" is not clear. Specifically, how does the generative model constrain the solution set?
>
> By optimizing the latent variable (instead of the pixels) it is ensured that the reconstructed image is within the range of the generator G. Thus the solution set is restricted to those images that can be generated by G. We clarified this in the updated abstract.
>
> > In 2.1. B. Back-projection section, why this serves as a regularizer for the solution by step A? Isn't z_0 in this section \bar{z_0}?
>
> By back-projecting we search in the a latent variable that is close to the optimized intermediate representation $z_i^*$. Since we are assuming a standard normal latent variable, the ball constraint around 0 ensures that we do not search in regions with low probability. We updated the notation in the paper to stress this.
>
> > The reviewer encourages to empirically compare the "random restarts and to select the solution with the lowest magnitude error after the optimization" (in Sec. 2.2) to the proposed method.
> Likely in Figure 4. Also citation is needed for this sentence.
>
> We added a citation for this sentence. The requested analysis is indeed shown in Figure 4. In Section 2.2 we introduce the initialization schemes. In Section 3.3 we describe the experiments in broader detail.
>
> > Please explain why you only evaluate LI initialization on Fourier phase retrieval.
>
> LI requires training an encoder network that maps from the measurements to the corresponding latent variable. For the Gaussian phase retrieval problem the measurement process changes for each m. Thus the encoder network needs to be retrained for each m, which we found a bit impractical for potential users. However, we expect improved performance for applying LI for Gaussian phase retrieval as well. This matter is explained in the paragraph below Equation (12) on page 8.
>
> > Please explain why why step C (the proposed method) is not very effective in StyleGAN on CelebA for Fourier measurements (Table 3 - PRILO-LI section)
>
> Note, that our results are better and Table 3 demonstrates that the naive application of ILO is not fruitful as the results of "only step A" are better than "only step A and B" (which corresponds to naively applying the approach by Daras et al. [6]). Keep in mind, that PSNR values are on a logarithmic scale.
>
> > In Table 1, the latest method compared is DPR [12], which is published in 2018. We have list of relevant work as follows to name a few: [...]
>
> Thanks for pointing these works out! We were not aware of them as they are very recent. Please note, that the works of Chen et al. are not applicable to our problem as they only consider the non-oversampled Fourier phase retrieval problem and since they use untrained neural networks (which are weaker priors than trained generative models) we do not expect any useful results.
>
> We argue that the provided baselines (DPR+DCGAN, DPR-MII+DCGAN, DPR+Progessive GAN, DPR-MII+Progressive-GAN, DPR-MII+StyleGAN and DPR+StyleGAN) **are the most appropriate baselines to verify our claims.** However, we contacted Killedar et al. and asked for the code.
>
> > In all equation reference, "Equation equation ??" should be "Equation ??"
>
> We fixed this in the current revision. Thanks for spotting this error.
>
> > Figure 1 in text (C. Refinement in Sec. 2.1) has no link.
>
> Thanks for pointing this out. We fixed it in the current revision.

---

> > ### Comment · Reviewer_J5zR · 2022-10-30
> > **Follow up questions**
> >
> > > While analysis of the retrieved phases could be interesting, in practical applications one is usually only interested in reconstructing the images, e.g. electron densities.
> >
> > $\to$ I think it should be important to demonstrate how your proposed method works in phase retrieval even though magnitude visualization is practically important.
> >
> > > The Fourier phase retrieval problem is the problem that is more relevant in practice and more difficult to solve (in the non-oversampled case). Compressive Gaussian phase retrieval is a theoretical toy problem that we included for completeness and to compare against DPR.
> >
> > $\to$ Then, lower your voice in addressing Gaussian phase retrieval (e.g., additionally we address Gaussian...) as you didn't fully demonstrate the benefit of your method in Gaussian.
> >
> > > By optimizing the latent variable (instead of the pixels) it is ensured that the reconstructed image is within the range of the generator G. Thus the solution set is restricted to those images that can be generated by G. We clarified this in the updated abstract.
> >
> > $\to$ Do you have any empirical results to support this claim?
> >
> > > We added a citation for this sentence. The requested analysis is indeed shown in Figure 4. In Section 2.2 we introduce the initialization schemes. In Section 3.3 we describe the experiments in broader detail.
> >
> > $\to$ In Fig. 4, which line is for "random restarts and to select the solution with the lowest magnitude error after the optimization"?
> >
> > > Note, that our results are better and Table 3 demonstrates that the naive application of ILO is not fruitful as the results of "only step A" are better than "only step A and B" (which corresponds to naively applying the approach by Daras et al. [6]). Keep in mind, that PSNR values are on a logarithmic scale.
> >
> > $\to$ My question is to ask explaining why the gain in PRILO-LI is smaller than PRILO-MII.
> >
> > > We argue that the provided baselines (DPR+DCGAN, DPR-MII+DCGAN, DPR+Progessive GAN, DPR-MII+Progressive-GAN, DPR-MII+StyleGAN and DPR+StyleGAN) are the most appropriate baselines to verify our claims. However, we contacted Killedar et al. and asked for the code.
> >
> > $\to$ Then, do you get the code?

---

> > > ### Author Response · Authors · 2022-11-02
> > > **Response to follow up questions by reviewer J5zR**
> > >
> > > > I think it should be important to demonstrate how your proposed method works in phase retrieval even though magnitude visualization is practically important.
> > >
> > > There might be some confusion about the task we have solved: phase retrieval asks to reconstruct an image $x$ from magnitude measurements of the Fourier transform of the image $y = |F(x)|$. In Figure 2 we show the reconstructed images (in practical applications these images are for example electron densities). From these images one could again calculate the Fourier magnitude and phase. We do not visualize the phase and magnitudes as there is nothing exciting to observe. Reconstructing the images is the goal of this work.
> > >
> > > > Then, lower your voice in addressing Gaussian phase retrieval (e.g., additionally we address Gaussian...) as you didn't fully demonstrate the benefit of your method in Gaussian.
> > >
> > > Ok, we rephrased the abstract accordingly. Please, see the last sentence of the new revision.
> > >
> > > > Do you have any empirical results to support this claim?
> > >
> > > This is a well-known phenomenon and thoroughly discussed in Section 6.3.2 (and Figure 5-7) of Bora et al. (https://arxiv.org/pdf/1703.03208.pdf). Note, that Bora et al. use the same setup we are also considering in our experiments, i.e., a DCGAN trained on CelebA data.
> > >
> > > > In Fig. 4, which line is for "random restarts and to select the solution with the lowest magnitude error after the optimization"?
> > >
> > > All lines use the "random restarts and to select the solution with the lowest magnitude error after the optimization" (number of restarts is on the x-axis). The lines differ in the used initialization (PRILO = random sample from the latent variable, PRILO-LI = learned initialization with additional perturbation as described in second paragraph on page 6, PRILO-MII = magnitude informed initialization.)
> > >
> > > > My question is to ask explaining why the gain in PRILO-LI is smaller than PRILO-MII.
> > >
> > > Intuitively, the closer the initial latent code predicted by the encoder gets to the optimal solution the less likely it is to get stuck in local optima. In comparison to PRILO-MII the learned initialization PRILO-LI provide a significantly better initial reconstruction before any optimization is done. Therefore, there are less possibilities to stop in local minima and the gain is smaller.
> > >
> > > > Then, do you get the code?
> > >
> > > Unfortunately, the authors did not reply to our mail. That is why we spent an effort to reimplement the paper: Please refer to https://anonymous.4open.science/r/sdlss-FD3E/SDLSS.ipynb. However, we were not able to reproduce the results at all (as you can see in the last cell of the Jupyter notebook).

---

> > > > ### Comment · Reviewer_J5zR · 2022-11-06
> > > > **Thank you for the clarification.**
> > > >
> > > > Thank you!

---

### Comment · Action_Editors · 2022-09-22
**Delay**

Dear Authors,

Sorry that there was a delay with one of the reviews.  I expect we will proceed to the next stage by the end of the weekend.

Area Chair

---

### Decision · Action_Editors · 2022-11-07

**Recommendation:** Accept as is

**Comment:**

I judge that the paper satisfies the main two criteria of (i) supporting the claims made, and (ii) being of interest to some TMLR readers.  The reviewers all opted for 'leaning accept', so none were strongly in favor, but at least all were in favor.  Some concerns remained about reproducibility and lack of realistic phase retrieval applications, but I don't view these as deal-breakers.

**Audience:**

This is a good match for TMLR.  Inverse problems with deep generative models are of broad interest to multiple communities.  This paper gives sufficient new findings/ideas that some individuals will be interested.

**Claims And Evidence:**

The main claims are copied from the paper as follows:
Claim 1: We claim that the idea of optimizing intermediate representations in generative models can be applied to solve phase retrieval problems. However, given the difficulty of these non-linear inverse problems an additional subsequent optimization step is required to obtain good results.
Claim 2: We also claim that the need for multiple runs with random restarts for generator-based approaches to solve inverse problems can be reduced by using different initialization schemes

The authors propose recovery algorithms and initialisation schemes to support these claims, and illustrate them with a variety of numerical experiments.  As one reviewer notes, perhaps the main limitation is that it's questionable whether these scenarios (image recovery with an accurate generator) would arise in typical applications of phase retrieval.  But the paper seems to be sufficiently clear about the assumptions and limitations.

---

> ### Author Response · Authors · 2022-11-17
> **Camera-ready version**
>
> Dear all,
>
> we have uploaded the camera-ready version. We would like to thank everyone involved for their time, their insightful comments and the fast turnaround. Thank you!
>
> Tobias Uelwer, Sebastian Konietzny, Stefan Harmeling